# RNA-Seq Bulked Segregant Analysis of an Exotic *B. napus* ssp. *napobrassica* (Rutabaga) F_2_ Population Reveals Novel QTLs for Breeding Clubroot-Resistant Canola

**DOI:** 10.3390/ijms25094596

**Published:** 2024-04-23

**Authors:** Zhiyu Yu, Rudolph Fredua-Agyeman, Stephen E. Strelkov, Sheau-Fang Hwang

**Affiliations:** Department of Agricultural, Food and Nutritional Science, University of Alberta, Edmonton, AB T6G 2P5, Canada; zhiyu3@ualberta.ca (Z.Y.); sheau.fang.hwang@ualberta.ca (S.-F.H.)

**Keywords:** *Brassica napus*, clubroot resistance, BSR-seq, QTLs, SNPs, *Plasmodiophora brassicae*

## Abstract

In this study, a rutabaga (*Brassica napus* ssp. *napobrassica*) donor parent FGRA106, which exhibited broad-spectrum resistance to 17 isolates representing 16 pathotypes of *Plasmodiophora brassicae*, was used in genetic crosses with the susceptible spring-type canola (*B. napus* ssp. *napus*) accession FG769. The F_2_ plants derived from a clubroot-resistant F_1_ plant were screened against three *P. brassicae* isolates representing pathotypes 3A, 3D, and 3H. Chi-square (χ^2^) goodness-of-fit tests indicated that the F_2_ plants inherited two major clubroot resistance genes from the CR donor FGRA106. The total RNA from plants resistant (R) and susceptible (S) to each pathotype were pooled and subjected to bulked segregant RNA-sequencing (BSR-Seq). The analysis of gene expression profiles identified 431, 67, and 98 differentially expressed genes (DEGs) between the R and S bulks. The variant calling method indicated a total of 12 (7 major + 5 minor) QTLs across seven chromosomes. The seven major QTLs included: *BnaA5P3A.CRX1.1*, *BnaC1P3H.CRX1.2*, and *BnaC7P3A.CRX1.1* on chromosomes A05, C01, and C07, respectively; and *BnaA8P3D.CRX1.1*, *BnaA8P3D.RCr91.2*/*BnaA8P3H.RCr91.2*, *BnaA8P3H.Crr11.3*/*BnaA8P3D.Crr11.3*, and *BnaA8P3D.qBrCR381.4* on chromosome A08. A total of 16 of the DEGs were located in the major QTL regions, 13 of which were on chromosome C07. The molecular data suggested that clubroot resistance in FGRA106 may be controlled by major and minor genes on both the A and C genomes, which are deployed in different combinations to confer resistance to the different isolates. This study provides valuable germplasm for the breeding of clubroot-resistant *B. napus* cultivars in Western Canada.

## 1. Introduction

Clubroot, caused by *Plasmodiophora brassicae* Woronin, is an important soilborne disease of cruciferous crops worldwide [1,2]. Infection by *P. brassicae* results in excessive growth and division of the host root cells, resulting in the formation of root galls and an eventual reduction in the plant’s capacity for water and nutrient uptake [3,4]. The cruciferous genus *Brassica* is known for its economically important agricultural and horticultural crops [5]. These include Chinese cabbage, turnip, Polish canola, and other crops belonging to the species *Brassica rapa* (A genome); cabbage, cauliflower, broccoli, kale, Brussels sprouts, and others classified as *B. oleracea* (C genome), and rutabaga and canola/oilseed rape, which are *B. napus* (AC genome) [1,5,6]. Globally, average yield losses caused by clubroot are estimated at 10% to 15% but may be as high as to 30% to 100% under favourable conditions [1,7,8]. The clubroot pathogen survives as resting spores that can persist in the soil for many years, making the management of this disease difficult [9,10].

In Alberta and other Canadian provinces, clubroot has emerged as a constraint to canola (*B. napus* var. *napus* L.) production [11,12,13]. The number of *P. brassicae*-infested fields in Alberta has increased from 12 in 2003 [14] to 3894 individual fields by 2022 [15]. Although clubroot-resistant canola varieties represent the most effective and environmentally friendly strategy for clubroot management [13,16], *P. brassicae* populations show high diversity in terms of virulence and can quickly adapt to overcome host resistance [11,12,17]. Over the past decade, ‘resistance-breaking’ pathotypes have been documented in hundreds of fields across Alberta [11,12,18]. Forty-three pathotypes of *P. brassicae*, as classified on the Canadian Clubroot Differential (CCD) set [11], have been reported to date from Canadian collections of the pathogen [12]. A majority of these pathotypes are highly virulent on canola cultivars carrying ‘first-generation’-type resistance [12], which appears to be derived from the European oilseed rape cv. ‘Mendel’ [19].

Genetic mapping is important for the identification of clubroot resistance (CR) gene loci and for the development of molecular markers for marker-assisted selection (MAS). Conventional PCR-based markers, such as amplified fragment length polymorphisms (AFLPs), cleaved amplified polymorphic sequences (CAPSs), random amplification of polymorphic DNA (RAPD), restriction fragment length polymorphisms (RFLPs), sequence characterized amplified regions (SCARs), sequence tagged sites (STSs), and simple sequence repeats (SSRs), were used widely for linkage-based identification and mapping clubroot-resistance gene loci before the era of sequencing technologies [20,21,22,23,24,25,26]. Next-generation sequencing (NGS) for genetic mapping has facilitated the development and application of genomics tools in plant breeding, such as genotyping-by-sequencing, single-nucleotide polymorphism (SNP) arrays, and bulked segregant analysis (BSA) [27]. Bulked segregant RNA sequencing (BSR-seq) is one of the most cost-effective methods for mapping genes of interest via BSA. This process evaluates two bulk DNAs or RNAs of plants with different phenotypes for differentially expressed genes (DEGs) and maps QTLs by variant calling [28]. Multiple CR gene loci against Canadian pathotypes of *P. brassicae*, such as *Rcr1*, *Rcr2*, *Rcr3*, *Rcr6* and *Rcr9^wa^*, have been identified via BSA or BSR-seq [29,30,31,32].

Rutabaga (*B. napus* ssp. *napobrassica*) could be a good source of clubroot resistance genes for emerging virulent *P. brassicae* pathotypes. Old rutabaga varieties like ‘Wilhelmsburger’ are known to carry resistance effective against most Canadian *P. brassicae* pathotypes [11,33]. Hasan and Rahman [34] found a Canadian rutabaga cv. ‘Brookfield’, which was resistant to all five ‘old’ pathotypes (2F, 3H, 5I, 6M, and 8N) found in Canada prior to the introduction of clubroot-resistant canola. Wang et al. [35] used a rutabaga cv. ‘Polycross’ as a resistance donor to breed for canola populations resistant to three Canadian pathotypes. Fredua-Agyeman et al. [21] observed that 87.9% of 124 rutabaga accessions from Nordic countries showed resistance to at least one of 16 Canadian *P. brassicae* pathotypes.

In this study, an F_2_ population derived from rutabaga accession FGRA106 that was reported to be resistant to 17 isolates representing 16 pathotypes of *P. brassicae* [21] was evaluated for its reaction to pathotypes 3A, 3D, and 3H. The inheritance of the resistance was determined based on segregation ratios. The genomic regions that co-segregated with resistance were determined based on the number of differentially expressed genes (DEGs), and the quantitative trait loci (QTLs) were mapped by BSR-seq.

## 2. Results

### 2.1. Clubroot Tests

The chi-square tests of homogeneity indicated that the phenotypic data of F_2_ plants inoculated with all three pathotypes were not significantly different in the three replicates (Appendix A). Therefore, data for the same pathotype were pooled for analysis. The frequency distribution of disease ratings to the three pathotypes is presented in Figure 1. The inoculation conducted on the F_2_ plants with *P. brassicae* pathotype 3A showed that 12.3%, 7.4%, 32.1%, and 48.3% (*n* = 408) of the plants were rated as 0, 1, 2, or 3, respectively (Appendix A). The screening results with pathotype 3D indicated that 29.4%, 11.4%, 17.8%, and 41.4% of the F_2_ plants (*n* = 411) exhibited disease ratings of 0, 1, 2, or 3, respectively (Appendix A). In response to *P. brassicae* pathotype 3H, 23.0%, 9.8%, 13.0%, and 54.2% of the F_2_ population showed disease ratings of 0, 1, 2, or 3 (*n* = 439) (Appendix A). Based on Fisher’s LSD test on the disease reaction data, the virulence of the pathotypes on the F_2_ population was in the order of 3A > 3H = 3D.

### 2.2. Inheritance of Clubroot Resistance in F_2_ Populations

The chi-square goodness-of-fit test was carried out in two ways following the protocol of Fredua-Agyeman et al. [37]. The first method grouped plants with a disease rating = 0 or 1 as resistant (R) and those with scores 2 or 3 as susceptible (S); meanwhile, the second method grouped plants with a disease rating = 0 as R and all others as S. The results obtained with the first method showed that the segregation of clubroot resistance in the F_2_ population was not significantly (*p* < 0.05) different from the expected Mendelian segregation ratios (R:S) of 3:13, 7:9, and 5:11 for pathotypes 3A, 3D, and 3H, respectively, all of which fit the two-gene models (Table 1). The second method indicated that the R:S ratios for pathotypes 3D and 3H were not significantly different from 5:11 (two-gene model) or 1:3 (one-gene model), respectively, while the ratio for pathotype 3A significantly deviated from all assumptions (Table 1).

### 2.3. RNA Sequencing, Filtering, and Sequence Alignment

The raw RNA sequences of the resistant and susceptible bulks were filtered, and the adapters were removed. Subsequently, the number of clean reads retained in the resistant bulks ranged from 24.2 to 28.7 Gb, 23.5 to 26.5 Gb, and 22.3 to 25.0 Gb, while in the susceptible bulks, this number ranged from 21.5 to 42.9 Gb, 20.7 to 23.2 Gb, and 21.8 to 23.8 Gb for pathotypes 3A, 3D, and 3H, respectively (Appendix A). Therefore, the RNA sequencing data yielded 20× to 30× the genome size of *B. napus*. The GC content ranged from 47% to 48%. Approximately 89.7% to 93.1% of these reads were mapped to the *B. napus* cv. ‘ZS11’ reference genome v2.0, 68.6% to 71.9% of which mapped only to exonic gene regions (Appendix A). The mismatch rate per base ranged from 0.9% to 1.1% (Appendix A). Therefore, the sequencing data were of adequate quality for the subsequent analysis.

### 2.4. SNP Calling and Marker Distribution 

A total of 338,177, 331,344, and 325,623 SNP markers were obtained for the comparisons between the reference (*B. napus* cv. ‘ZS11’) genome and the resistant and susceptible bulks from the inoculation experiments with pathotypes 3A, 3D, and 3H, respectively (Table 2 and Figure 2). The SNP marker densities on all 19 *B. napus* chromosomes are presented in Table 2 and Figure 2. The SNP densities for the pathotype 3A, 3D, and 3H bulks on the different chromosomes ranged from 190.78 to 732.31, 187.07 to 719.08, and 189.14 to 692.25 SNPs/Mb, respectively (Table 2). The highest SNP density was found on chromosome A10 for the three pathotypes, while the lowest occurred on chromosome C02 (Table 2). Consistently high SNP densities were observed at the beginning of chromosomes A01, A02, and A03, as well as at the end of chromosome A10, across all three pathotypes. Conversely, a region of low coverage density was noted on chromosome C09 (Figure 2). 

### 2.5. Differentially Expressed Genes

Totals of 73,607, 72,644, and 72,361 differentially expressed genes were identified between the resistant and susceptible bulks for pathotypes 3A, 3D and 3H, respectively (Figure 3). About 1.15% (850), 0.17% (120), and 0.21% (151) of these genes were significantly (*P_adj_* < 0.05) differentially expressed (Figure 3). Based on a 95% confidence threshold (*P_adj_* < 0.05) and using |log2 FC| > 2 as the criteria, 428, 67, and 98 DEGs were identified in the bulks of pathotypes 3A, 3D, and 3H (Appendix A, Figure 4). Among these, 81, 27, and 36 genes were upregulated in the R bulks of 3A, 3D, and 3H, respectively, while 347, 40, and 62 genes were upregulated in the S bulks (Figure 5a,b). One DEG was consistently identified in the R bulks, S bulks, and both R and S bulks. Two, nine and thirteen DEGs were found between the bulks of pathotypes 3A and 3D, 3D and 3H, and 3A and 3H, respectively. About 72% (412/569), 13% (76/569), and 10% (56/569) of the DEGs were detected for pathotypes 3A, 3H, and 3D, respectively (Figure 5c).

### 2.6. Identification of QTLs Associated with Clubroot Resistance

Based on Δ(SNP-index) statistics at a 99% confidence interval (CI) from the variant calling between the R and S bulks of the three pathotypes, a total of 12 QTLs associated with resistance to 3 three pathotypes were detected on 7 of the 19 chromosomes of *B. napus* (Figure 6, Appendix A). These QTLs were located on chromosomes A01 (1), A05 (1), A08 (4), C01 (3), C07 (2), C08 (1), and C09 (1) (Appendix A). The peak |Δ(SNP-index)| values ranged from ~0.23 to 0.53 (Appendix A). The higher the |Δ(SNP-index)|, the stronger the correlation between marker SNPs and traits. In this study, the QTLs were classified as major if the |Δ(SNP-index)| > 0.32 and the peak was clearly above the 99% confidence interval; conversely, QTLs were classified as minor if the |Δ(SNP-index)| < 0.30 and the peak fell between the 95% and 99% confidence intervals (Appendix A).

The QTLs were named following the *Brassica* gene nomenclature system proposed by Østergaard and King [39], as modified by Fredua-Agyeman et al. [37]. For example, the QTL on chromosome A01 was designated *BnaA1P3D.CRX1.1*, where the first letter denotes the genus (*Brassica*), the second and third letters the species (*napus*), the fourth letter the genome (*A*), the fifth letter the chromosome (*1*), and the sixth, seventh, and eighth letters (*P3D*) the pathotype of *P. brassicae* used for inoculation. These are followed by the name(s) of the closest published CR gene(s) (3–8 letters) or the letter *X* if no previous markers have been reported (*CRX*), and finally, the number of the QTL number (two digits, *1.1*). 

Based on this gene nomenclature, two major effect QTLs, *BnaA5P3A.CRX1.1* on chromosome A05 and *BnaC7P3A.CrrA51.1* on chromosome C07, conferred resistance to the isolate representing pathotype 3A (Figure 7; Table 3 and Appendix A). Four major and five minor QTLs were identified for resistance to the isolate representing pathotype 3D. The four major QTLs, *BnaA8P3D.CRX1.1*, *BnaA8P3D.RCr91.2*, *BnaA8P3D.Crr11.3*, and *BnaA8P3D.qBrCR381.4*, were all situated on chromosome A8. The five minor QTLs, *BnaA1P3D.CRX1.1, BnaA5P3D.CRX1.1*, *BnaC7P3D.CRX1.1*, *BnaC8P3D.CRX1.1*, and *BnaC9P3D.CRX1.1*, were located on chromosomes A01, A05, C07, C08, and C09, respectively (Figure 7; Table 3 and Appendix A). Seven QTLs (three major and four minor) conferred resistance to the isolate representing *P. brassicae* pathotype 3H. Two of the major QTLs *BnaA8P3H.RCr91.2* and *BnaA8P3H.Crr11.3*, along with one minor QTL *BnaA8P3H.qBrCR381.4*, were located on chromosome A08. Additionally, one major QTL *BnaC1P3H.CRX1.2* and two minor QTLs *BnaC1P3H.CRX1.1* and *BnaC1P3H.CRX1.3* were found on chromosome C01. One minor QTL, *BnaC9P3H.CRX1.1*, was located on chromosome C09 (Figure 7; Table 3 and Appendix A).

QTLs located within 2 cM (~1000 kb) of each other were regarded as coincident and treated as the same QTL. Four of these coincident genomic regions provided resistance to two pathotypes. The coincident QTLs *BnaA5P3A.CRX1.1* (12,272,166 nt)/*BnaA5P3D.CRX1.1* (13,623,271 nt) on chromosome A05 conferred resistance to isolates representing pathotypes 3A and 3D. Similarly, the coincident QTLs *BnaA8P3D.RCr91.2* (10,275,090 nt)/*BnaA8P3H.RCr91.2* (11,310,428 nt) and *BnaA8P3H.Crr11.3* (15,908,289 nt)/*BnaA8P3D.Crr11.3* (16,016,864 nt) on chromosome A08, along with *BnaC9P3D.CRX1.1*/*BnaC9P3H.CRX1.1* (31,610,782 nt) on chromosome C09, conferred resistance to isolates representing pathotypes 3D and 3H (Figure 7; Table 3 and Appendix A).

### 2.7. Genes Identified in Clubroot Resistance QTL Regions

A total of 11 of the 12 identified QTL regions contained genes with the exception of *BnaC8P3D.CRX1.1* (Table 3 and Appendix A). These included genes involved in the plant disease response such as ethylene-responsive transcription factor ERF109 (gene ID: 106361033), serine/threonine-related genes (106418155), heat shock protein (HSP) genes (gene ID: 106361308, 106362025), polyubiquitin 11 (gene ID: 106349084), GLABROUS1 enhancer-binding protein-like genes (gene ID: 106376056), UDP-glucosyltransferase genes (gene ID: 106361045), and 60S ribosomal protein genes (gene ID: 106406248, 111202359).

### 2.8. DEGs Identified in Clubroot Resistance QTL Regions

Seventeen annotated DEGs were found in four of the identified QTLs regions associated with clubroot resistance. Thirteen of these genes were located on chromosome C07, while three genes were on chromosome A08 (Table 4). Eight of the DEGs (IDs: 106348481, 106348998, 106390302, 106410495, 106410578, 106410663, 106410664, 111198409), all on chromosome C07, were identified in the hosts inoculated with the isolate representing pathotype 3A. One of these genes (ID# 106348481) encoded DOWNY MILDEW RESISTANCE 6 protein (Table 4). Two genes, IDs 106407096 and 106348764 on chromosome C07, were expressed in the reactions to pathotype 3D. The latter of these, which encoded tesmin/TSO1-like CXC 7, was also differentially expressed in response to pathotype 3H (Table 4). Three other genes on chromosome C07 (IDs: 106349452, 106440113, 111204564) were also expressed following inoculation with the isolate representing pathotype 3H. Another three genes on chromosome A08 (IDs: 106360694, 106381656, 106416269) were also upregulated in response to pathotype 3H (Table 4). Other genes in the QTL region associated with the plant disease response included ethylene-responsive transcription factor 1A-like (gene ID: 106381656) on *BnaA8P3D.RCr91.2*, 60S ribosomal protein L7-2 (gene ID: 106349452) on *BnaC7P3D.CRX1.1*, and heat stress transcription factor A-7a-like (gene ID: 111198409) on *BnaC7P3A.CRX1.1*.

### 2.9. Functional Enrichment Analyses of Differentially Expressed Genes

The Gene Ontology (GO) analysis indicated a significant enrichment (FDR < 0.05) of GO terms for pathotypes 3A, 3D, and 3H, with 19, 22, and 59 enriched terms, respectively (see Appendix A). Figure 8 illustrates the top GO terms with the highest number of enriched DEGs. The results showed that DEGs in the pathotype 3A bulks were involved in bioprocesses previously reported to induce or enhance plant immunity, such as response to sucrose (GO:0009744), aminoglycan metabolism (GO:0006022) and catabolism (GO:0006026), chitin metabolism (GO:0006030) and catabolism (GO:0006032), and defense response (GO:0006952) [40,41,42,43]. The DEGs in the pathotype 3D bulks were involved in methionine regulation-related bioprocesses (GO:0019509, GO:0071267, GO:0071265, GO:0009086, GO:0006555), actin filament bundle assembly (GO:0051017) and organization (GO:0061572), and sulfur amino acid biosynthetic (GO:0000097) and metabolic process (GO:0000096). The DEGs in the pathotype 3H bulks were involved in tRNA or mitochondrial tRNA-related processes (GO:0034414, GO:0042779, GO:0042780, GO:0072684, GO:0000963, GO:0000959, GO:1905267), inositol phosphate-related processes (GO:0046855, GO:0071545, GO:0046854, GO:0043647, GO:0046856, GO:0006661), phospholipid biosynthetic (GO:0008654), metabolic (GO:0006644) and dephosphorylation processes (GO:0046839), and chitin metabolism (GO:0006030) and catabolism (GO:0006032). 

The KEGG pathway analyses indicated that 14, 7, and 10 pathways were significantly (FDR < 0.05) associated with the DEGs in response to pathotypes 3A, 3D, and 3H, respectively (Appendix A, Figure 9). Metabolic (bna01100) and ribosome (bna03010) pathways were associated with DEGs for all three pathotypes. In contrast, the plant–pathogen interaction (bna04626) and glycine, serine, and threonine metabolism (bna00260) pathways were associated only with DEGs for pathotype 3A. Glutathione metabolism (bna00480) was linked to the DEGs in response to both pathotypes 3A and 3H. Phenylpropanoid (bna00940) and flavonoid (bna00941) biosynthesis were associated with DEGs for pathotypes 3D and 3H.

## 3. Discussion

The German rutabaga cv. ‘Wilhelmsburger’ (ECD 10) was originally proposed as a differential host by Williams [33] and subsequently included in both the European Clubroot Differential (ECD; [44]) and CCD [11] sets. However, Yu et al. [22] and Fredua-Agyeman et al. [21] observed different resistance phenotypes in seven ‘Wilhemsburger’ accessions from Denmark, FGRA106, FGRA107, FGRA108, FGRA109, FGRA110, FGRA111, and FGRA112, when these were challenged with the same set of isolates representing 16 different *P. brassicae* pathotypes from Canada. Only the ‘Wilhemsburger’ accession FGRA106 from Denmark showed broad-spectrum clubroot resistance comparable with that of ‘Wilhelmsburger’ (ECD 10) from Germany; both were resistant to the 16 pathotypes tested by Fredua-Agyeman et al. [21]. Given the proximity of Denmark to Germany as neighboring countries and the potential for germplasm movement, it is plausible that the ‘Wilhelmsburger’ accession FGRA106, based on its reactions, could be equivalent to ‘ECD 10’ and thus might harbor the same CR gene(s).

‘Wilhelmsburger’ has been used as a clubroot resistance donor in breeding programs worldwide for many decades [45]. Lammerink [46] evaluated F_2_ progenies derived from ‘Wilhelmsburger’ with a *P. brassicae* isolate designated ‘Race B’ and suggested that the resistance was controlled by one dominant gene based on a 3R:1S segregation ratio. Similarly, Ayers and Lelacheur [47] reported that, based on segregation ratios of an F_2_ population derived from ‘Wilhelmsburger’, the resistance to *P. brassicae* race 2 (*sensu* Williams, 1966 [33]) was controlled by two dominant genes, whereas resistance to race 3 was controlled by one dominant gene. In a separate study involving an F_2_ population, Gustafsson and Falt [48] reported that the resistance of ‘Wilhelmsburger’ to a less virulent isolate ‘Pb3’ may have been conferred by two genes, whereas resistance to a highly virulent isolate ‘Pb7’ appeared to involve only one gene. In contrast, Crute et al. [49] suggested that ‘Wilhelmsburger’ possesses three clubroot resistance genes. These observations indicate that ‘Wilhelmsburger’ may carry multiple CR genes that could be differentially effective depending on the virulence of specific *P. brassicae* isolates. In the current study, the segregation ratios of F_2_ plants inoculated with Canadian *P. brassicae* isolates representing various pathotypes were analyzed. The results suggested that the resistance inherited from FGRA106 to pathotypes 3A and 3D was likely determined by two genes, whereas resistance to pathotype 3H was conferred by either one or two genes. 

Crute et al. [49] suggested that the CR genes in *B. rapa* (A genome) and *B. napus* (AC genome) are qualitative or race-specific, while resistance in *B. oleracea* (C genome) is quantitative or race-nonspecific [49,50]. In this study, two major QTLs, *BnaA5P3A.CRX1.1* on chromosome A05 and *BnaC7P3A.CRX1.1* on chromosome C07, conferred resistance to an isolate representing pathotype 3A. The QTL *BnaC7P3A.CRX1.1* were mapped to a position proximal to the *CrrA5* gene on chromosome A05 [51], while the QTL *BnaC7P3A.CRX1.1* were mapped distal to the *Rcr7*, *qCRc7-1*, *qCRc7-2*, *qCRc7-3*, and *qCRc7-4* genes or QTLs on chromosome C07 [52,53] (Figure 10). This indicates the possibility of novel CR genes in these regions. In the case of the isolates representing pathotypes 3D and 3H, the major QTLs *BnaA8P3D.Rcr91.2* and *BnaA8P3H.Rcr91.2* were mapped to the same genomic location as *Rcr1* [54,55], while *BnaA8P3H.Crr11.3* and *BnaA8P3D.Crr11.3* were mapped to the same genomic location as *Crr1* [20,56] (Figure 10). Another major QTL, *BnaA8P3D.qBrCR381.4*, detected in response to pathotype 3D, was mapped to the same genomic location on chromosome A08 as *qBrCR38-2* [32,57], while the major QTL *BnaA8P3D.CRX1.1* was mapped to a position proximal to the aforementioned genes on this chromosome (Figure 10). This suggests that two additional genomic regions on chromosome A08 were needed to confer resistance to pathotype 3D.

Chromosome A08 has been reported to harbour major CR loci in rutabaga [21]. Four major QTLs in this study were positioned on the A08 chromosome. Despite the differences in mapping methods and reference genomes, the identified QTLs were located in genomic regions where *Crr1* [20,56], *CRs* [57], *PbBa8.1* [56], *Rcr9* [55], *Rcr9^wa^* [32], and *Rcr9^ECD01^* [54] were previously mapped. The *Crr1* gene was identified from progenies of *B. rapa* ‘Siloga’ and conferred resistance to a Japanese isolate of *P. brassicae* classified as pathotype/race 4 [20]. The gene *CRs* [57] was mapped in inbred lines derived from an unknown CR turnip donor, which provided resistance to pathotype 4 [33]; meanwhile, *Rcr9* [55], conferring resistance to pathotype 5X [11], was derived from a German turnip cultivar ‘Pluto’. Additionally, *Rcr9^wa^* or *PbBa8.1*, which confers resistance to pathotype 5X [11] or pathotypes 4 and 7 [33], was inherited from *B. rapa* ECD 04 [32,56]. The gene *Rcr9^ECD01^*, derived from *B. rapa* ECD 01, provided resistance to pathotypes 3A, 3D, 3H, and 5X, which includes the three pathotypes examined in this study [54]. 

Two major effect QTLs on the C genome were mapped to chromosomes C01 and C07. The first major QTL, *BnaC1P3H.CRX1.2*, which conferred resistance to pathotype 3H, was distant from the QTL *Rcr_C01-1* reported on chromosome C01 of *B. oleracea* [58] (Figure 10). Two minor effect QTLs, *BnaC1P3H.CRX1.1* and *BnaC1P3H.CRX1.3* on chromosome C01, which also conferred resistance to pathotype 3H, were identified as proximal and distal, respectively, to a major effect QTL *BnaC1P3H.CRX1.2*. The second major effect QTL, *BnaC7P3A.CRX1.1*, which conferred resistance to pathotype 3A, was located on the bottom half of chromosome C07. A minor QTL, *BnaC7P3D.CRX1.1*, was also located on the bottom half of chromosome C07. These results confirm the bottom half of chromosome C07 as a genomic hotspot for several clubroot resistance genes, including *Rcr7*, *qCRc7-1*, *qCRc7-2*, *qCRc7-3*, and *qCRc7-4* [52,53]. Therefore, the clubroot resistance derived from the donor FGRA106 in this study appears to be conferred not only by major QTLs on the A genome, but also by major and minor QTLs located on the C genome.

The plant immune system is generally considered to comprise two layers: pathogen-associated molecular patterns (PAMPs)-triggered immunity (PTI), which offers basal protection against many pathogens, and effector-triggered immunity (ETI), which results in robust and localized responses against specific pathogens [59]. The GO and KEGG enrichment analyses conducted in this study suggested that the DEGs between the R and S bulks were associated with both of these layers of defense. The GO analysis revealed the differential expression of genes involved in bioprocesses related to sucrose, actin filaments, and sulfur amino acids, while enriched KEGG pathways were associated with metabolic pathways, ribosomes, and tRNA. These processes and pathways have been implicated as common initial defense signaling processes in eukaryotes [60,61,62,63,64]. Additionally, pathways involving inositol phosphate, glycine, serine, and threonine were also identified and have been implicated in the recognition of PAMPs by pattern recognition receptors (PRRs) in the host and in subsequent defense responses [60,65,66,67,68]. Several GO processes and KEGG pathways were also identified that have been implicated in ETI, including GO bioprocesses related to methionine, phospholipid and chitin and KEGG pathways associated with glutathione, phenylpropanoid, and flavonoids [42,62,69,70,71,72]. 

Some of the genes identified in the QTL regions through the SNPs and differential gene expression analyses have also been associated with PTI and ETI. For instance, heat shock proteins (HSPs) function as chaperones, playing roles in protein folding, assembly, translocation, and degradation during both abiotic and biotic stress. These processes are vital for the formation of PRRs and intracellular responsive proteins essential for resistance [73]. Polyubiquitin modulates cellular protein turnover and homeostasis in basal host defense to abiotic and biotic stresses, and it is involved in the responsive modification of proteins in both PTI and ETI [74]. The ethylene-responsive transcription factor ERF109 plays a key role in ethylene-mediated defense pathways during ETI [75]. GLABROUS1 enhancer-binding protein-like, UDP-glucosyltransferase, and 60S ribosome proteins are implicated in ETI, and silencing of these genes can activate plant defense pathways [76,77,78,79]. Additionally, downy mildew resistance 6 serves as a resistance gene in ETI [80]. In a recent study of differential gene expression in ‘Wilhelmsburger’ in response to inoculation with pathotype 3A, Zhou et al. [81] found that salicylic acid and ethylene-mediated defense were involved in the host reaction. 

## 4. Materials and Methods

### 4.1. Plant Materials

The parental materials consisted of the clubroot-resistant *B. napus* ssp. *napobrassica* accession FGRA106 (identified as the cultivar ‘Wilhemsburger’) [21,22] and the susceptible *B. napus* ssp. *napus* accession FG769 (spring canola cv. ‘Sedo’) [82]. FGRA106 was reported to be resistant (disease severity index (DSI) ≤ 30%) to isolates of *P. brassicae* representing pathotypes 2F, 5I, 6M, 5X (LG-2), 5L, 2B, 3A, 8E, 5K, 3O, and 8P and moderately resistant (30% < DSI ≤ 50%) to pathotypes 3H, 5X (LG-1), 8N, 5C, 5G, and 8J [21,22]. Genetic crosses were carried out by emasculation followed by hand pollination as follows: FGRA106 (♂) × FG769 (♀). An F_1_ hybrid plant that was resistant to *P. brassicae* pathotype 3H was vernalized for 10 weeks at 4 °C under a 12 h photoperiod and self-pollinated to obtain F_2_ seeds.

### 4.2. Phenotyping Assays

The parents and F_2_ population were screened against one single-spore isolate each of *P. brassicae* pathotypes 3A and 3H and a field isolate of pathotype 3D, as classified based on the CCD set [11]. A total of 1620 F_2_ individuals were tested in three rounds of bioassays, in which 180 plants were inoculated with each pathotype in each experiment. The inoculations were conducted as described previously by Strelkov et al. [3,83] with slight modifications. Briefly, clubbed roots were blended in sterile water and filtered through two layers of cheesecloth to generate a resting spore suspension. The spore concentration was then adjusted to 1 × 10^7^ resting spores/mL with sterile water. For inoculation, the roots of 7-day-old seedlings were dipped into the spore suspension for about 10 s and then planted in plastic pots (6 cm × 6 cm × 6 cm) filled with Sunshine Mix #4 potting mixture (Sun Gro Horticulture, Seba Beach, AB, Canada) at a density of one seedling per pot. One millilitre of the resting spore suspension was then pipetted in the potting mix around each seedling to ensure infection and minimize disease escape. In addition to the resistant and susceptible parents FGRA106 and FG769, respectively, the susceptible *B*. *napus* cv. ‘Westar’ was also included as a positive control in all experiments. 

The inoculated plants were kept in a greenhouse maintained at 25 °C/18 °C day/night with a 16 h light period (natural light supplemented with artificial lighting) and assessed for clubroot symptoms at 7 weeks after inoculation (wai) on a 0–3 disease severity scale as described by Kuginuki et al. [36] and Strelkov et al. [3], where: 0 = no galling, 1 = slight galling on side roots, 2 = moderate galling on main and side roots, and 3 = severe galling with almost no observable side roots. 

### 4.3. Bulk Construction and RNA Extraction

RNA extraction from the R (resistant) and S (susceptible) plant pools was based on the phenotypic reactions of individual plants to the three pathotypes. For each pathotype, 15 plants with a disease rating of 0 were pooled into an R bulk, while each S bulk consisted of 15 plants with a rating of 2 or 3. Three biological replicates of both the R and S bulks were assigned for each pathotype. 

Leaf samples of each bulk collected at 7 weeks after inoculation were mixed and ground into powder in liquid nitrogen. The total RNA from each bulk replicate was extracted from 0.1 mL (~100 mg) of powdered root tissue of each sample using an RNeasy Plant Mini Kit (Qiagen; Toronto, ON, Canada) and purified using an RNase-Free Dnase kit (Qiagen; Toronto, ON, Canada). The RNA concentration was measured with a NanoDrop 2000c spectrophotometer (Thermo Scientific; Waltham, MA, USA), and its quality (RNA integrity numbers (RIN) ≥ 6.5 and a 28S/18S ratio ≥ 1.0) was confirmed using an Agilent 2200 TapeStation system (Agilent, Santa Clara, CA, USA).

### 4.4. RNA Sequencing

The cDNA library preparation and RNA sequencing were performed by the Oklahoma Medical Research Foundation NGS Core (Oklahoma City, OK, USA) with an IDT xGen RNA Library kit (Integrated DNA Technologies, Inc., San Diego, CA, USA) and an Illumina NovaSeq 6000 S4 platform (Illumina; San Diego, CA, USA). Pair-end read sequences (2 × 150 bp) were generated in ‘fastq’ format for further analysis.

### 4.5. Sequence Alignment, Identification of DEGs, and Variant Calling

The adaptors were removed from the raw sequences using GATK v4.2.2.0 [84] and quality checked with FastQC v0.12.1 (https://www.bioinformatics.babraham.ac.uk/projects/fastqc/, accessed on 15 November 2023). The trimmed sequences were then aligned to the *B. napus* cv. ‘ZS11’ reference genome v2.0 (https://www.ncbi.nlm.nih.gov/datasets/genome/GCF_000686985.2/, accessed on 15 November 2023) using STAR v2.7.9a [85]. Three R and three S bulks of each pathotype were analyzed as replicates to identify differentially expressed genes (DEGs), and they were pooled for variant calling. Gene expression read counts were calculated using RSEM v1.3.3 [86] and normalized with the R package ‘DESeq2’ [87]. The significance of differentially expressed genes (DEGs) between the R and S bulks was determined based on the log2 fold change (|log2 FC| > 2) for the bulk pairs of each pathotype. Volcano plots of DEG counts were generated using the R package ‘ViDGER’ [88]. Enrichment analyses of the Gene Ontology (GO; http://geneontology.org/, accessed on 15 December 2023) and Kyoto Encyclopedia of Genes and Genomes (KEGG; https://www.genome.jp/kegg/, accessed on 15 December 2023) databases were conducted with an eggNOG-mapper v2 [89] and KOBAS-i tool [90] for annotating disease-related biological processes and pathways.

The variant calling was performed using the GATK v4.2.2.0 function ‘HaplotypeCaller’, with the SNPs detected being subsequently filtered using the GATK ‘VariantFiltration’ function under proper standards (‘QD < 2.0||MQ < 40.0||FS > 60.0||SOR > 3.0’). The final *.vcf* files were converted to *.table* format using the GATK ‘VariantsToTable’ tool for analysis in R 4.2.1 [84,91]. SNP frequency calling was carried out on resistant and susceptible bulks as described by Liu et al. [28], Yu et al. [29], and Wu et al. [92] with modifications. Comparisons of SNP variants between the bulks were performed with the R package ‘QTLseqr’ [93] with the following filter settings: refAlleleFreq = 0.20, minTotalDepth = 50, maxTotalDepth = 500, minSampleDepth = 80, minGQ = 99. The detection of QTLs was based on the SNP-index and the ∆(SNP-index) in a 1 Mb sliding window [38]. The SNP-index statistic calculates marker association differences in the genotype frequencies of mixed pools, where a value of 0 indicates that the short reads contain genomic fragments from the reference parent, while a value of 1 indicates that all of the short reads represent the genome from the other parent [38]. A Δ(SNP-index) graph was used to detect the differences between the ‘highest’ and ‘lowest’ pools of extreme phenotypes, where Δ(SNP-index) = 1 and −1 indicates bulk DNA from one parent and the other parent, respectively, while Δ(SNP-index) = 0 if both parents have the same SNP-indices at the genomic regions [38]. The physical positions of the detected QTLs were visualized with MapChart v2.3.2 [94] and compared with previously reported QTLs for clubroot resistance.

### 4.6. Statistical Analyses

The phenotypic data from different replicates of the same inoculum (pathotype) were subjected to chi-square tests of homogeneity. A chi-square goodness-of-fit test was performed to determine the segregation of the phenotypic data for all three pathotypes using R 4.2.1. Other built-in R packages including ggplot2, reshape2, and ggrepel were also used for data analysis or visualization.

## 5. Conclusions

The CR donor FGRA106 and the resistant F_2_ progeny evaluated in this study were confirmed to carry resistance to *P. brassicae* pathotypes 3A, 3D, and 3H, which are predominant in canola in Western Canada [12]. The resistance donor FGRA106 exhibited reactions similar to ECD10 and was previously reported to be resistant or moderately resistant to 17 isolates representing 16 pathotypes of *P. brassicae* [21]. Based on the DEGs, QTLs, and associated GO terms and KEGG pathways, gene loci conferring resistance to pathotype 3A were mapped to chromosomes A05 and C07, while major QTLs for resistance to pathotypes 3D and 3H co-segregated to at least three genomic regions on chromosome A08. Another major QTL on chromosome C01 was required for resistance to pathotype 3H. The CR donor and the SNP markers identified in this study may serve as valuable resources for clubroot resistance breeding in Canadian canola.

## Figures and Tables

**Figure 1 ijms-25-04596-f001:**
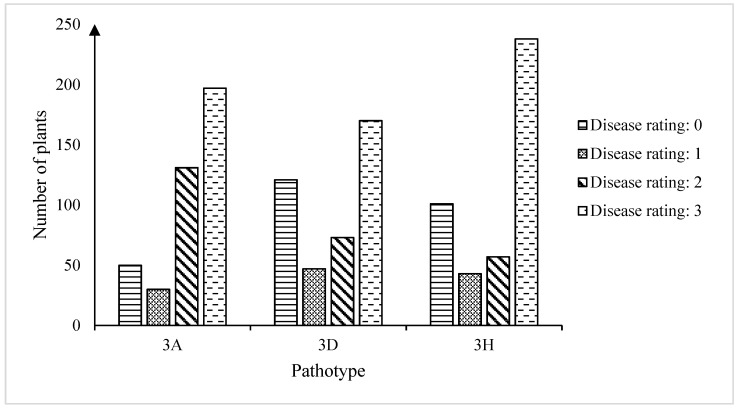
Frequency distribution of clubroot disease severity ratings in an F_2_ population derived from FGRA106 (♀) × FG769 (♂) to isolates representing pathotypes 3A, 3D, and 3H of *Plasmodiophora brassicae*. Plants were grown under greenhouse conditions and evaluated for clubroot severity on a 0–3 disease severity scale as described by Kuginuki et al. [36] and Strelkov et al. [3] at 7 weeks following inoculation with each pathotype.

**Figure 2 ijms-25-04596-f002:**
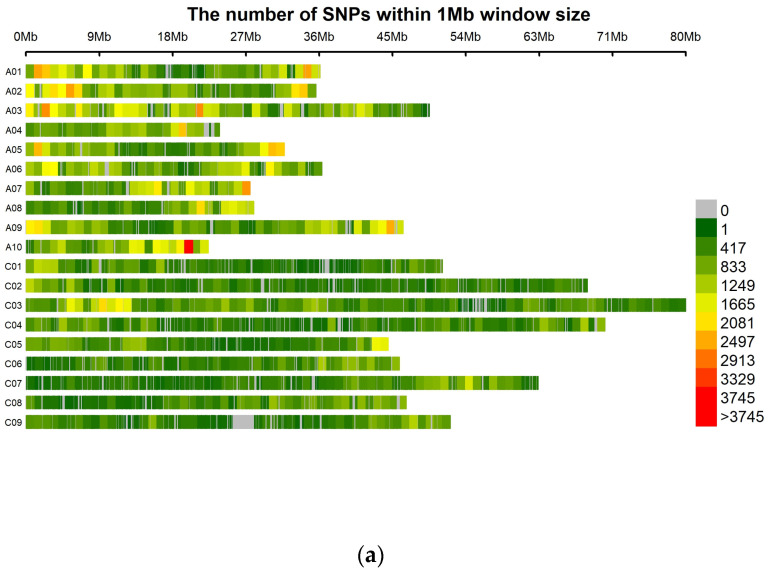
Distribution of polymorphic single-nucleotide polymorphisms (SNPs) on 19 *Brassica napus* chromosomes identified between resistant (R) and susceptible (S) bulks in an F_2_ population derived from FGRA106 (♀) × FG769 (♂) and tested with *Plasmodiophora brassicae* isolates representing pathotypes 3A (**a**), 3D (**b**), and 3H (**c**). The colors indicate SNP density (SNPs/Mb) as per the scale on the right-hand side.

**Figure 3 ijms-25-04596-f003:**
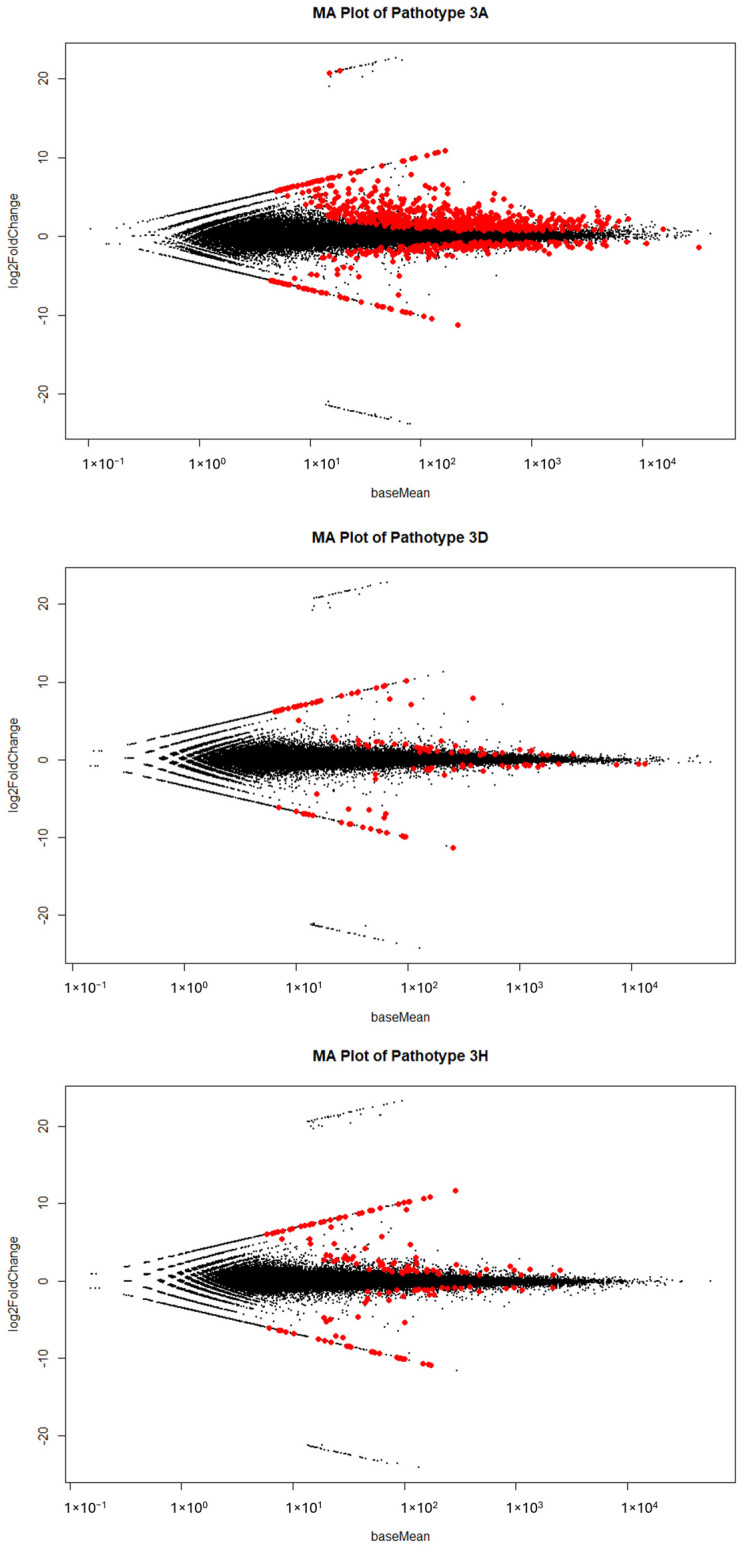
MA plots from base means (*x*−axis; ‘M’) and the average of log fold changes (*y*−axis; ‘A’), indicating differentially expressed genes in resistant (R) and susceptible (S) bulks of F_2_ plants derived from FGRA106 (♀) × FG769 (♂) and tested with isolates representing *P. brassicae* pathotypes 3A (top), 3D (middle), and 3H (bottom). Red spots indicate genes with *P_adj_* < 0.05. Black spots indicate genes with *P_adj_* ≥ 0.05.

**Figure 4 ijms-25-04596-f004:**
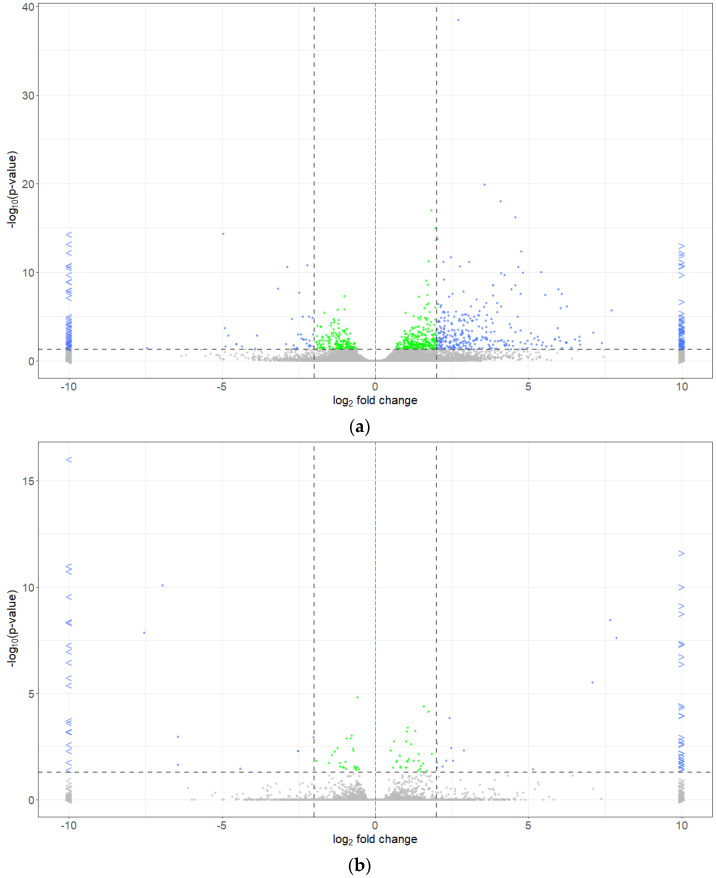
Volcano plots from the log2 fold change (*x*−axis) and −log10 *P_adj_* (*y*−axis) values for the selection of 428, 67, and 98 differentially expressed genes (DEGs) in resistant (R) or susceptible (S) bulks of F_2_ plants derived from FGRA106 (♀) × FG769 (♂) and tested with *Plasmodiophora brassicae* pathotypes 3A (**a**), 3D (**b**), and 3H (**c**), respectively. The log2 fold change indicates the mean expression level for each gene; each dot/arrowhead represents one gene. Grey dots and arrowheads: genes of *P_adj_* > 0.05; green dots: genes of *P_adj_* < 0.05 and |log2 fold change| < 2; blue dots and arrowheads: genes of *P_adj_* < 0.05 and |log2 fold change| > 2. Red dashed line: log2 fold change = 2 separating upregulated genes in the R and S bulks; dark gray dashed lines: thresholds of *P_adj_* = 0.05 (horizontal) and log2 fold change = 2 or −2 (vertical). The criteria for selection of DEGs in this study were *P_adj_* < 0.05 and |log2 fold change| > 2.

**Figure 5 ijms-25-04596-f005:**
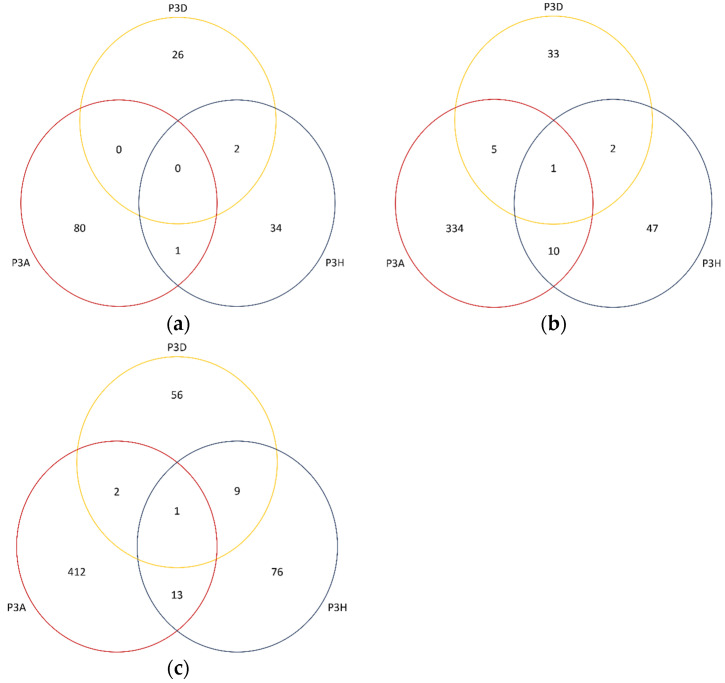
Venn diagrams indicating the overlaps of differentially expressed genes among resistant (**a**), susceptible (**b**), and all (**c**) bulks of F_2_ plants derived from FGRA106 (♀) × FG769 (♂) and tested with *Plasmodiophora brassicae* pathotypes. P3A, P3D, and P3H denote pathotypes 3A, 3D, and 3H, respectively.

**Figure 6 ijms-25-04596-f006:**
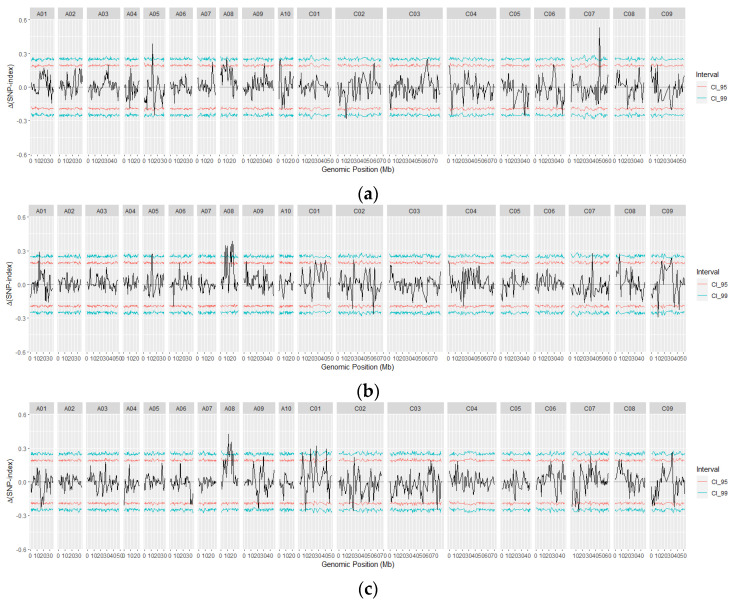
The variant calling of polymorphic single-nucleotide polymorphisms (SNPs) differing between clubroot-resistant (R) and susceptible (S) bulks of F_2_ plants derived from FGRA106 (♀) × FG769 (♂) and tested with *Plasmodiophora brassicae* pathotypes 3A (**a**), 3D (**b**), and 3H (**c**) on 19 *B. napus* chromosomes based on Δ(SNP-index) statistics (Takagi et al. [38]) at a 99% confidence interval. The *x*−axis indicates the position on the chromosomes, and the *y*−axis denotes the Δ(SNP-index), where CI_95 = 95% confidence interval and CI_99 = 99% confidence interval.

**Figure 7 ijms-25-04596-f007:**
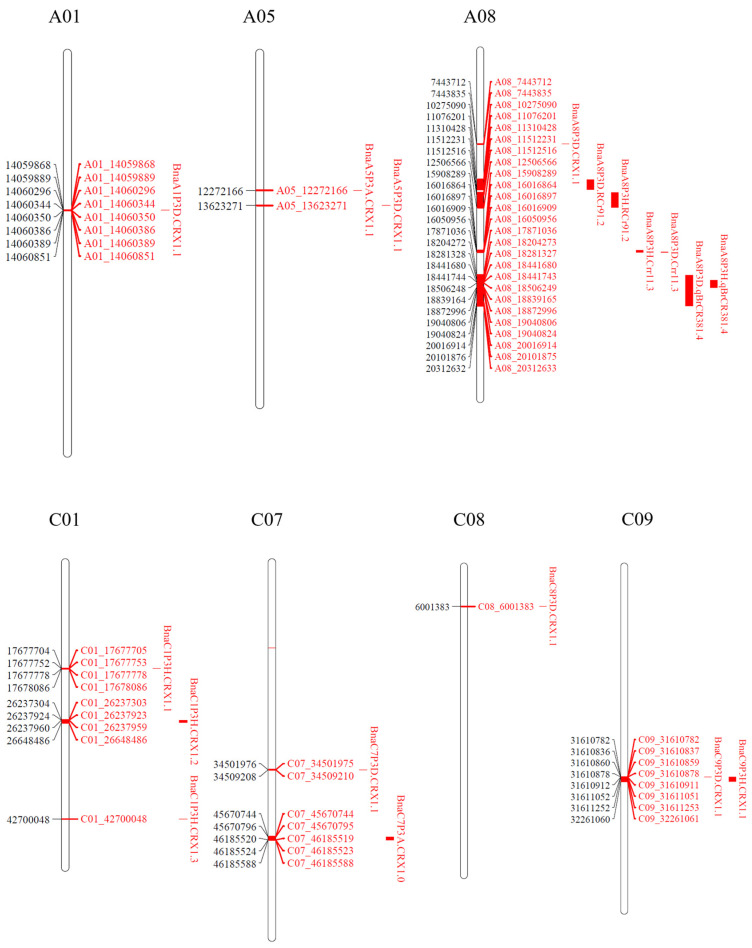
The single-nucleotide polymorphisms (SNPs) and QTLs identified from clubroot-resistant and susceptible bulks of F_2_ plants inoculated with *Plasmodiophora brassicae* pathotypes 3A, 3D, and 3H on chromosomes A01, A05, A08, C01, C07, C08, and C09. The positions and names of SNPs are denoted on each chromosome (left and right, respectively), while the flanking regions of identified QTLs are represented as red bars, both on and to the right of each chromosome.

**Figure 8 ijms-25-04596-f008:**
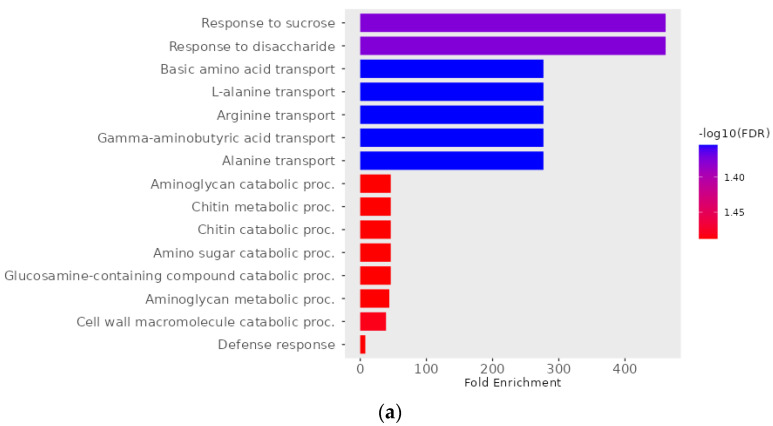
GO enrichment with differentially expressed genes between clubroot-resistant and susceptible bulks of F_2_ plants derived from FGRA106 (♀) × FG769 (♂) and tested with *Plasmodiophora brassicae* pathotypes 3A (**a**), 3D (**b**), and 3H (**c**). Visualization was based on the GO biological processes sorted by fold enrichment. The colors indicate a −log10 false discovery rate (FDR) as per the scale on the right-hand side.

**Figure 9 ijms-25-04596-f009:**
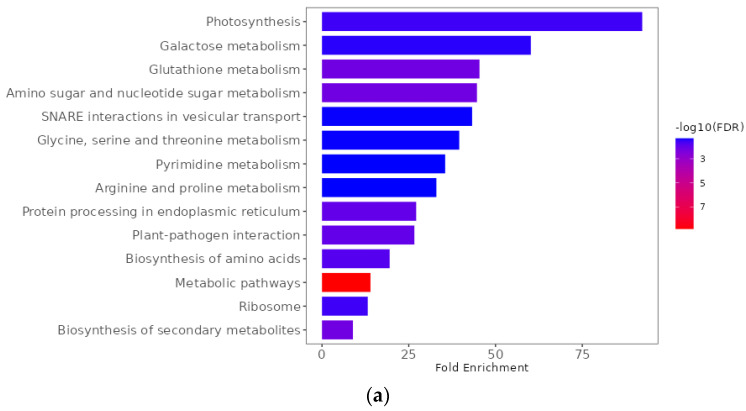
KEGG pathways associated with differentially expressed genes between clubroot-resistant and susceptible bulks of F_2_ plants derived from FGRA106 (♀) × FG769 (♂) and tested with *Plasmodiophora brassicae* pathotypes 3A (**a**), 3D (**b**), and 3H (**c**). Visualization of pathways was sorted by fold enrichment. The colors indicate a −log10 false discovery rate (FDR) as per the scale on the right-hand side.

**Figure 10 ijms-25-04596-f010:**
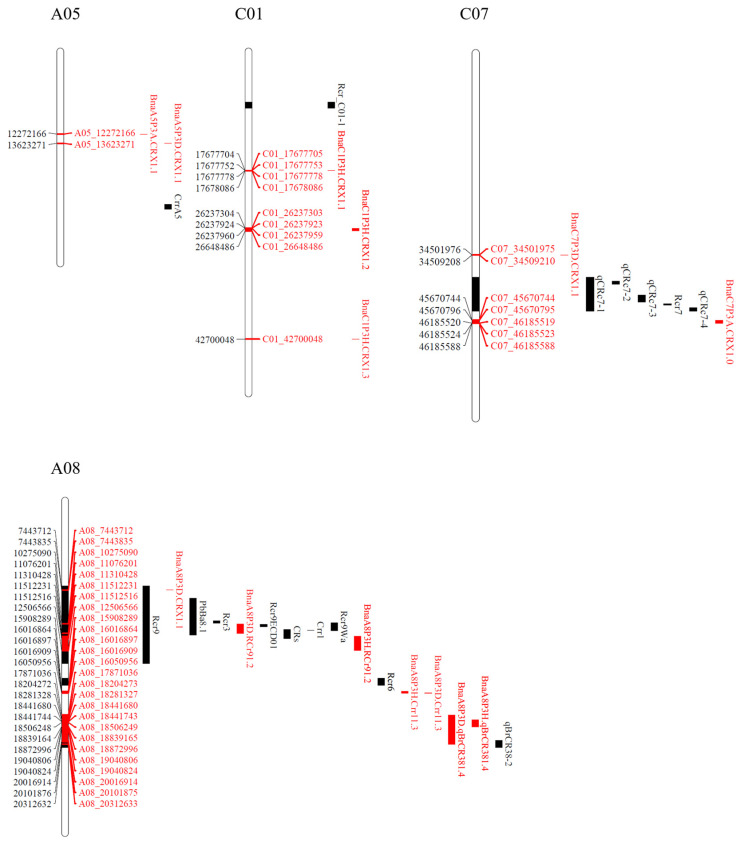
Comparison between previously reported clubroot-resistant (CR) gene loci and QTLs identified in this study on chromosomes A05, A08, C01, and C07. The position and name of SNPs identified in this study are indicated on each chromosome (left and right, respectively), and the flanking regions of previously reported CR gene loci and QTLs identified in this study are denoted by black and red bars, respectively, both on and to the right of each chromosome.

**Table 1 ijms-25-04596-t001:** Segregation ratios of resistant (R) and susceptible (S) plants in an F_2_ population derived from FGRA106 (♀) × FG769 (♂) evaluated for resistance to *Plasmodiophora brassicae* pathotypes 3A, 3D, and 3H.

Pathotype	Total	R (0 + 1)	S (2 + 3)	R:S Ratio ^1^	Chi-Square	*p*-Value	R (0)	S (1 + 2 + 3)	R:S Ratio ^2^	Chi-Square	*p*-Value
3A	408	80	328	3R:1S	667.6601	<0.00001	50	358	3R:1S	856.6797	<0.00001
				1R:3S	6.326797	0.01189			1R:3S	35.34641	<0.00001
				9R:7S	222.5982	<0.00001			9R:7S	320.8988	<0.00001
				7R:9S	96.62994	<0.00001			7R:9S	164.4544	<0.00001
				5R:11S	25.73975	<0.00001			5R:11S	68.5205	<0.00001
				11R:5S	458.6125	<0.00001			11R:5S	606.1205	<0.00001
				13R:3S	1017.633	<0.00001			13R:3S	1274.888	<0.00001
				3R:13S	0.197084	0.65708			3R:13S	11.29814	0.00078
				15R:1S	3827.712	<0.00001			15R:1S	4624.575	<0.00001
				1R:15S	124.2458	<0.00001			1R:15S	25.1085	<0.00001
3D	411	168	243	3R:1S	255.2482	<0.00001	121	290	3R:1S	454.9886	<0.00001
				1R:3S	55.24818	<0.00001			1R:3S	4.321979	0.03762
				9R:7S	39.4748	<0.00001			9R:7S	120.039	<0.00001
				7R:9S	1.379562	0.24018			7R:9S	34.1977	<0.00001
				5R:11S	17.72568	0.00003			5R:11S	0.626454	0.42866
				11R:5S	148.6348	<0.00001			11R:5S	295.6083	<0.00001
				13R:3S	439.767	<0.00001			13R:3S	724.1655	<0.00001
				3R:13S	132.0747	<0.00001			3R:13S	30.83218	<0.00001
				15R:1S	1960.994	<0.00001			15R:1S	2900.964	<0.00001
				1R:15S	840.9942	<0.00001			1R:15S	377.2303	<0.00001
3H	439	144	295	3R:1S	416.918	<0.00001	101	338	3R:1S	632.9301	<0.00001
				1R:3S	14.25133	0.00016			1R:3S	0.930144	0.33483
				9R:7S	98.08038	<0.00001			9R:7S	197.1373	<0.00001
				7R:9S	21.38196	<0.00001			7R:9S	76.7563	<0.00001
				5R:11S	0.492069	0.48301			5R:11S	13.88449	0.00019
				11R:5S	264.0557	<0.00001			11R:5S	427.5572	<0.00001
				13R:3S	676.3863	<0.00001			13R:3S	977.5294	<0.00001
				3R:13S	56.89907	<0.00001			3R:13S	5.221716	0.02231
				15R:1S	2783.138	<0.00001			15R:1S	3749.576	<0.00001
				1R:15S	528.2043	<0.00001			1R:15S	210.3765	<0.00001

^1^ Plants with clubroot disease severity ratings of 0 and 1 were regarded as resistant (R), and those with ratings of 2 and 3 as susceptible (S). ^2^ Plants with a clubroot disease severing rating of 0 were regarded as R, and those with ratings of 1, 2, and 3 as S.

**Table 2 ijms-25-04596-t002:** Distribution and density of single-nucleotide polymorphisms (SNPs) identified in resistant (R) and susceptible (S) bulks in an F_2_ population derived from FGRA106 (♀) × FG769 (♂) tested with pathotypes 3A, 3D, or 3H of *Plasmodiophora brassicae*.

Pathotype	3A		3D		3H	
Chromosome	# SNP ^1^	SNP/Mb ^2^	# SNP	SNP/Mb	# SNP	SNP/Mb
A01	20,939	584.52	20,597	574.97	19,994	558.14
A02	22,146	626.78	21,668	613.25	21,502	608.56
A03	31,343	637.84	30,494	620.56	29,862	607.70
A04	12,572	533.69	12,345	524.05	12,040	511.11
A05	16,995	539.98	16,512	524.63	16,373	520.22
A06	20,115	557.50	19,809	549.02	19,356	536.46
A07	15,927	580.71	15,526	566.09	15,439	562.91
A08	11,782	424.70	11,737	423.08	11,348	409.06
A09	23,060	501.92	22,594	491.78	22,382	487.16
A10	16,270	732.31	15,976	719.08	15,380	692.25
C01	11,826	232.90	11,388	224.28	11,236	221.28
C02	13,049	190.78	12,795	187.07	12,937	189.15
C03	26,556	330.38	26,055	324.15	25,843	321.51
C04	16,072	227.80	16,025	227.14	15,571	220.70
C05	12,100	274.10	11,924	270.11	11,722	265.54
C06	11,473	252.12	11,417	250.89	11,086	243.62
C07	13,456	215.49	12,956	207.49	12,986	207.97
C08	13,499	291.26	13,316	287.31	12,989	280.26
C09	10,939	211.59	10,503	203.16	10,074	194.86
Scaffolds	18,058	NA	17,707	NA	17,503	NA

^1^ # SNP, SNP count located on A and C chromosomes or scaffolds. ^2^ SNP/Mb, SNP density per million base pairs. NA, not applicable.

**Table 3 ijms-25-04596-t003:** QTLs conferring resistance to pathotypes 3A, 3D, or 3H of *Plasmodiophora brassicae* and genes identified by significant single-nucleotide polymorphisms (SNPs) in QTLs.

QTL	Chromosome	Position		Gene ID	Gene Name	Overlapping QTL
		Start	End			
*BnaA5P3A.CRX1.1* (Major)	A05	12272166	12272166	106362025	*hsp70-Hsp90 organizing protein 3*	
*BnaC7P3A.CRX1.1*	C07	45670744	46185588	106406248	*60S ribosomal protein L26-1*	
(Major)				111208271	*alpha-humulene/(-)-(E)-beta-caryophyllene synthase-like*	
*BnaA1P3D.CRX1.1* (Minor)	A01	14059868	14060851	106361131	*cilia- and flagella-associated protein 251*	
*BnaA5P3D.CRX1.1* (Minor)	A05	13623271	13623271	106411554	*proteasome subunit alpha type-4-A-like*	
*BnaA8P3D.CRX1.1* (Major)	A08	7443712	7443835	106396583	*enoyl-CoA delta isomerase 2, peroxisomal*	
*BnaA8P3D.RCr91.2* (Major)	A08	10275090	11076201	106418916	*transcription initiation factor TFIID subunit 1*	
*BnaA8P3D.Crr11.3*	A08	16016864	16050956	106361045	*UDP-glucosyl transferase 73B2-like*	*BnaA8P3H.Crr11.3*
(Major)				106361049	*polyubiquitin 11*	*BnaA8P3H.Crr11.3*
				106361048	*polyadenylate-binding protein 2*	*BnaA8P3H.Crr11.3*
*BnaA8P3D.qBrCR381.4* (Major)	A08	17871036	20312633	106361356	*probable beta-1,3-galactosyltransferase 4*	
				106361316	*31 kDa ribonucleoprotein, chloroplastic*	
				106361308	*heat shock 70 kDa protein 6, chloroplastic-like*	*BnaA8P3H.qBrCR381.4*
				106361304	*3-oxo-Delta(4,5)-steroid 5-beta-reductase-like*	*BnaA8P3H.qBrCR381.4*
				106361295	*aconitate hydratase 1*	
				106361282	*phosphoserine aminotransferase 1, chloroplastic*	*BnaA8P3H.qBrCR381.4*
				106361274	*50S ribosomal protein L25-like*	
				106405582	*protein PLASTID MOVEMENT IMPAIRED 1*	*BnaA8P3H.qBrCR381.4*
				106361258	*bZIP transcription factor 60*	*BnaA8P3H.qBrCR381.4*
				106451550	*GLABROUS1 enhancer-binding protein-like 1*	
				106384864	*uncharacterized LOC106384864*	
				106384865	*uncharacterized LOC106384865*	
				106361389	*NAD(P)H-quinone oxidoreductase subunit M, chloroplastic*	
				106361360	*apoptotic chromatin condensation inducer in the nucleus*	
*BnaC7P3D.CRX1.1* (Minor)	C07	34501975	34509210	106418155	*serine/threonine-protein kinase prp4*	
				106418157	*uncharacterized LOC106418157*	
*BnaC9P3D.CRX1.1* (Minor)	C09	31610782	31611253	106418720	*vicilin-like seed storage protein At2g18540*	*BnaC9P3H.CRX1.1*
*BnaA8P3H.RCr91.2* (Major)	A08	11310428	12506566	106390924	*selenium-binding protein 1*	
				106397231	*UDP-glycosyltransferase 75C1-like*	
				106422372	*5-amino-6-(5-phospho-D-ribitylamino)uracil phosphatase, chloroplastic-like*	
*BnaA8P3H.Crr11.3*	A08	15908289	16050956	106361033	*ethylene-responsive transcription factor ERF109*	
(Major)				106361045	*UDP-glucosyl transferase 73B2-like*	*BnaA8P3D.Crr11.3*
				106361049	*polyubiquitin 11*	*BnaA8P3D.Crr11.3*
				106361048	*polyadenylate-binding protein 2*	*BnaA8P3D.Crr11.3*
*BnaA8P3H.qBrCR381.4*	A08	18281327	18872996	106361308	*heat shock 70 kDa protein 6, chloroplastic-like*	*BnaA8P3D.qBrCR381.4*
(Major)				106361304	*3-oxo-Delta(4,5)-steroid 5-beta-reductase-like*	*BnaA8P3D.qBrCR381.4*
				106361282	*phosphoserine aminotransferase 1, chloroplastic*	*BnaA8P3D.qBrCR381.4*
				106405582	*protein PLASTID MOVEMENT IMPAIRED 1*	*BnaA8P3D.qBrCR381.4*
				106361258	*bZIP transcription factor 60*	*BnaA8P3D.qBrCR381.4*
*BnaC1P3H.CRX1.1* (Minor)	C01	17677705	17678086	106376056	*GLABROUS1 enhancer-binding protein-like*	
*BnaC1P3H.CRX1.2* (Major)	C01	26237303	26648486	106349084	*polyubiquitin 11*	
				106349049	*uncharacterized LOC106349049*	
*BnaC1P3H.CRX1.3* (Minor)	C01	42700048	42700048	111202359	*60S ribosomal protein L27-3*	
*BnaC9P3H.CRX1.1* (Minor)	C09	31610782	32261061	106418720	*vicilin-like seed storage protein At2g18540*	*BnaC9P3D.CRX1.1*
				106392952	*uncharacterized LOC106392952*	

Note: The QTLs are denoted as major or minor in parentheses below each QTL name based on peak |Δ(SNP-index)| values. Gene IDs and names were obtained from the NCBI database (https://www.ncbi.nlm.nih.gov/, accessed on 1 December 2023), where gene names denote descriptions of gene functions. Overlapping QTLs denote the same QTL identified in bulks tested with different pathotypes.

**Table 4 ijms-25-04596-t004:** Differentially expressed genes (DEGs) in identified QTL regions of resistant (R) and susceptible (S) bulks in an F_2_ population derived from FGRA106 (♀) × FG769 (♂) tested for resistance to *Plasmodiophora brassicae* pathotypes 3A, 3D and 3H.

Gene ID	Symbol	Chromosome	QTL	DEG Pathotype	Description of Gene Functions (Gene Name)
106360694	LOC106360694	A08	*BnaA8P3D.RCr91.2*	3H	*RGG repeats nuclear RNA binding protein A*
106381656	LOC106381656	A08	*BnaA8P3D.RCr91.2*	3H	*ethylene-responsive transcription factor 1A-like*
106416269	LOC106416269	A08	*BnaA8P3D.qBrCR381.4*	3H	*cysteine protease XCP2*
106407096	LOC106407096	C07	*BnaC7P3A.CRX1.1*	3D	*SMAX1-LIKE 3 protein*
106348481	LOC106348481	C07	*BnaC7P3A.CRX1.1*	3A	*downy mildew resistance 6 protein*
106348764	LOC106348764	C07	*BnaC7P3A.CRX1.1*	3D, 3H	*tesmin/TSO1-like CXC 7 protein*
106348998	LOC106348998	C07	*BnaC7P3A.CRX1.1*	3A	*peptidyl-prolyl cis-trans isomerase FKBP65*
106349452	LOC106349452	C07	*BnaC7P3D.CRX1.1*	3H	*60S ribosomal protein L7-2*
106390302	LOC106390302	C07	*BnaC7P3D.CRX1.1*	3A	*LOB domain-containing protein 37-like*
106410495	LOC106410495	C07	*BnaC7P3A.CRX1.1*	3A	*glycine-rich protein 5-like*
106410578	LOC106410578	C07	*BnaC7P3A.CRX1.1*	3A	*pectinesterase inhibitor 7-like*
106410663	LOC106410663	C07	*BnaC7P3A.CRX1.1*	3A	*bark storage protein A*
106410664	LOC106410664	C07	*BnaC7P3D.CRX1.1*	3A	*tonoplast dicarboxylate transporter*
106440113	LOC106440113	C07	*BnaC7P3A.CRX1.1*	3H	*protein CHAPERONE-LIKE PROTEIN OF POR1, chloroplastic*
111198409	LOC111198409	C07	*BnaC7P3A.CRX1.1*	3A	*heat stress transcription factor A-7a-like*
111204564	LOC111204564	C07	*BnaC7P3A.CRX1.1*	3H	*ATP-dependent Clp protease ATP-binding subunit CLPT1, chloroplastic-like*

Note: Gene IDs, symbols, and names were obtained from the NCBI database (https://www.ncbi.nlm.nih.gov/, accessed on 1 December 2023), where gene names denote descriptions of gene functions.

## Data Availability

The data presented in the study are included in the article or as Appendix A. Further inquiries can be directed to the corresponding author.

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
