# Peer review of "RNA-Seq Bulked Segregant Analysis of an Exotic B. napus ssp. napobrassica (Rutabaga) F2 Population Reveals Novel QTLs for Breeding Clubroot-Resistant Canola"

_ijms, 2024, doi:10.3390/ijms25094596_

Round 1
Reviewer 1 Report
Comments and Suggestions for Authors
Yu and his/her colleagues reported the mapping of clubroot resistance QTLs in rutabaga using a F2 segregating population via BSA RNA-Seq method. Many candidate genes were identified within the QTLs by differentially expression analysis and gene set enrichment ananlysis. The QTLs will be valuable targets for disease resistance breeding and the candidate genes are good starts for gene cloning. The findings will be interesting to the readers of the journal. The study design is good and the results are reasonablly displayed. The organization of the manuscript is quite clear and easy to fellow. Minor revision is recommanded before consideration for publication.
minor concern:
1. line137-138: Based on the methods, GATK haplotypercaller was used for germline vairant calling. The resulting variants should be germline mutations compared to the reference genome, not the mutations between two bulks. So the relevant sentences could be revised.
2. Figure 7 is not very clear because of low resolution.
Reviewer 2 Report
Comments and Suggestions for Authors
The reviewed manuscript, titled "RNA-Seq bulked segregant analysis of an exotic B. napus ssp. napobrassica (rutabaga) F2 population reveals novel QTLs for breeding clubroot resistant canola" by Yu et al. is a fundamentally sound study that is focused on the identification and genetic characterization of features that are associated with resistance to the plant pathogen that causes clubroot. Overall this work is outstanding - the authors designed a solid set of experiments that ultimately culminated with the identification of several novel QTLs that are associated with resistance to this disease. Throughout the work, the authors clearly and deliberately presented and explained their rationale, their experiments, and the interpretations are clearly presented and supported by their data. I enjoyed reading this manuscript immensely, and i believe that many other researchers will do the same. I have a few suggestions, all are relatively minor in nature and I believe can be easily addressed during revision.
Major comments:
Line 126: The shift in writing and focus from section 2.2 to 2.3 is a bit abrupt. The previous section ends with an analysis of the statistical analysis of the resistant vs. susceptible plants into the # of clean reads. Please revise this for clarity and to connect these experiments in a manner that presents a more logical flow to this section.
Line 205: Section 2.6 contains a wealth of very relevant and highly interesting information. I would suggest that the authors compile a figure or a table of the QTLs for clarity and to make them easier for the reader to digest. A table would make the results easier visualize, and can be broken up into the major effect and minor effect QTLs, their genomic position, etc and then the text can be modified in this section to help the reader follow the data.
Table #3 and #4 are excellent. These happen to fall loosely under a region that is this reviewers field of study, which leads to this next suggestion, which I think would be a very important consideration for these authors. Namely, that there are a number of co-regulated DEGs that exhibit a layer of spatial positioning and clustering as a component of their normal regulation. A number of these clusters varies but they have been seen conserved across eukaryotes and involve many different processes - including defense mechanisms, secondary metabolite biosynthesis, and others. As a result of the non-random genomic distribution that is seen here, I encourage the authors to directly address this in regards to their DEGs that are reported on these tables. I would recommend the following reference:
"Functional clustering of metabolically related genes is conserved across Dikarya" by Cittadino et al. 2023
As an appropriate reference in this instance.
Minor notes:
There are a few typos and minor grammatical issues (see line 497 for typo example). Overall there are very few of these and the writing and clarity are exceptional. I recommend that the authors give the final revised manuscript one final read through for copy editing purposes prior to resubmission to resolve.
Some of the figures in the initial submission have fonts sizes, etc that are too small. This is common with the initial review copy of a manuscript, but please keep an eye on that in the page proof process to make sure that the fine work done here is easy to read.
Comments on the Quality of English LanguageThe language simply requires one final review prior to the submission of the final version.
